# Neuroprotective Effects of Sodium Butyrate by Restoring Gut Microbiota and Inhibiting TLR4 Signaling in Mice with MPTP-Induced Parkinson’s Disease

**DOI:** 10.3390/nu15040930

**Published:** 2023-02-13

**Authors:** Tong-Tong Guo, Zheng Zhang, Yan Sun, Rui-Yang Zhu, Fei-Xia Wang, Lian-Ju Ma, Lin Jiang, Han-Deng Liu

**Affiliations:** 1Laboratory of Tissue and Cell Biology, Experimental Teaching Center, Chongqing Medical University, Chongqing 400016, China; 2Department of Molecular Medicine and Cancer Research Center, Chongqing Medical University, Chongqing 400016, China

**Keywords:** sodium butyrate, microbiome–gut–brain axis, neuroinflammation, gut microbial dysbiosis, Parkinson’s disease

## Abstract

Parkinson’s disease (PD) is a prevalent type of neurodegenerative disease. There is mounting evidence that the gut microbiota is involved in the pathogenesis of PD. Sodium butyrate (NaB) can regulate gut microbiota and improve brain functioning in neurological disorders. Hence, we examined whether the neuroprotective function of NaB on PD was mediated by the modulation of gut microbial dysbiosis and revealed its possible mechanisms. Mice were administered 1-methyl-4-phenyl-1,2,3,6-tetrahydropyridine (MPTP) for 7 consecutive days to construct the PD model. NaB gavage was given 2 h after the daily MPTP injections for 21 days. NaB improved the motor functioning of PD mice, increased striatal neurotransmitter levels, and reduced the death of dopaminergic neurons. The 16S rRNA sequencing analysis revealed that NaB restored the gut microbial dysbiosis. NaB also attenuated the intestinal barrier’s disruption and reduced serum, colon, and striatal pro-inflammatory cytokines, along with inhibiting the overactivation of glial cells, suggesting an inhibitory effect on inflammation from NaB throughout the gut–brain axis of the PD mice. Mechanistic studies revealed that NaB treatment suppressed the TLR4/MyD88/NF-kB pathway in the colon and striatum. In summary, NaB had a neuroprotective impact on the PD mice, likely linked to its regulation of gut microbiota to inhibit gut–brain axis inflammation.

## 1. Introduction

Parkinson’s disease (PD) is a progressive neurodegenerative condition mainly affecting the elderly [1] and is mainly characterized by stiffness, resting tremors, bradykinesia, constipation, and depression [2]. Dopaminergic neuron loss in the substantia nigra along with a decline in dopamine (DA) released to the striatum constitutes the pathological PD hallmark [3]. Neuroinflammation is regarded as an important pathophysiological feature of PD, which is associated with neurodegeneration [4]. Although current treatments can alleviate PD symptoms, they are not able to prevent its neurodegeneration.

Benefiting from the accumulated research on the gut–brain axis, recent studies have demonstrated that PD pathogenesis may be affected by alterations in the gut microbiota [5,6,7]. Pathological changes in microbial taxonomy have been observed among PD patients. For example, the abundance of *Akkermansia*, *Lactobacillus*, and *Bacteroides* genera increased in PD patients, while that of *Prevotella*, *Faecalibacterium*, and *Roseburia* decreased remarkably [8,9,10,11]. Lin et al. reported that increased *Bacteroides* abundance in PD patients correlated positively with the severity of their motor disorders [9]. Interestingly, the feces of patients with PD transplanted in germ-free mice showed severe motor impairments, inflammation, and other pathological symptoms [12]. Additionally, Sun et al. found that transplanting healthy donor gut microbiota into mice with 1-methyl-4-phenyl-1,2,3,6-tetrahydropyridine (MPTP)-induced PD substantially improved the bodily injuries in these recipient mice and exerted a neuroprotective effect [13]. These studies indicate that the gut microbiota can be potentially targeted against PD. Remarkably, gut microbial dysbiosis not only leads to intestinal inflammation but also increases intestinal permeability. Harmful toxins and inflammatory factors from the intestinal tract enter the bloodstream via the damaged intestinal barrier and cross the blood–brain barrier, triggering or aggravating neuroinflammation, ultimately leading to dopaminergic neuronal death in patients with PD [14,15,16]. Hence, remodeling the gut microbiota to reduce intestinal permeability and inhibiting intestinal inflammation to reduce neuroinflammation and elicit neuroprotective effects in PD may be a promising treatment strategy for these patients.

Butyrate is a short-chain fatty acid (SCFA) synthesized by anaerobes in the colon of animals through the fermentation of indigestible dietary fibers [17]. Butyrate exerts anti-inflammatory and [18] anti-oxidant effects [19] while repairing the intestinal barrier [20]. Notably, butyrate also influences intestinal microbiota regulation. For example, Zhou et al. reported that sodium butyrate (NaB) reduced the entry of inflammatory factors into the liver by regulating gut microbial dysbiosis, thereby reducing steatohepatitis induced by a high-fat diet [21]. Furthermore, Ma et al. found that NaB regulated gut microbial composition, specifically reducing harmful bacteria and increasing beneficial bacteria, in mice with liver metastasis of colon cancer [22]. NaB can improve motor symptoms, neuronal loss, and neuroinflammation in MPTP-induced PD mice [23,24,25]. Nevertheless, the specific mechanism behind the neuroprotective properties of NaB against PD is mostly obscure. In our work, we speculated that NaB could exert a neuroprotective role in PD mice by altering the gut microbiota’s composition, thereby suppressing gut–brain axis inflammation.

We assessed the effects of NaB on motor symptoms, decline in striatal tyrosine hydroxylase (TH) levels, dopaminergic neuron loss, and reduction in striatal neurotransmitter expression in PD mice. Subsequently, we investigated the function of NaB in gut microbiota dysbiosis and regulating the intestinal barrier’s integrity in mice. Next, we examined the influence of NaB on inflammation across the gut–brain axis and neuroinflammation mediated by glial cells. Furthermore, we elucidated the potential molecular interactions between neuroinflammation and gut microbiota. Our results suggested that NaB prevented neurodegeneration in the PD mice by attenuating inflammation of the entire gut–brain axis, which might be achieved by targeting the gut microbiota, and also involved the TLR4/MyD88/NF-κB pathway.

## 2. Material and Methods

### 2.1. Animals and Treatment

Male C57BL/6 mice that were 7 weeks old were obtained from Changzhou Cavens Experimental Animal Co., Ltd. (Jiangsu, China) and housed for 7 days to acclimate to the standard habitat before being used in the experiment. Mice were kept in standard settings, including a 12 h dark/light cycle, 55 ± 10% humidity, and 25 ± 1 °C temperatures. Food and water were available to the mice without restriction. The Chongqing Medical University Ethics Committee granted its approval for all of the animal experiments.

Mice were randomly classified (*n* = 12 per group) into the three following groups: control; MPTP; and MPTP + NaB groups. To create the MPTP mouse model, we applied a technique that was previously reported [25,26]. Specifically, 30 mg/kg MPTP (Sigma Co., Ltd., St. Louis, MO, USA) was administered intraperitoneally to mice in the MPTP and MPTP + NaB groups for 7 consecutive days. In the mice belonging to the control group, an injection of an equivalent volume of normal saline solution was administered. NaB (Sigma Co., Ltd., USA) or saline solution was intragastrically administered, lasting for 14 days after 2 h of MPTP injection for 21 days. The experimental timeline is displayed in Figure 1A.

### 2.2. Behavioral Tests

#### 2.2.1. Pole Test

The climbing device used was a smooth wooden stick with a diameter and length of 1 cm and 50 cm, respectively. A small ball of 2 cm diameter was fixed at the top of the stick and wrapped in gauze to prevent slipping. On the wooden ball, the animal was placed upside down and the time to climb the complete pole from top to bottom was noted. A three-day training period preceded the start of the actual experiment. The average value of 3 measurements was calculated, and the interval between each measurement was 15 min.

#### 2.2.2. Rotarod Test

The animal was positioned on the rotarod, rotating at a rate of 4 rpm. The rotarod accelerated evenly at 40 rpm in 5 min; mouse fall time was also recorded. The animals were subjected to the rotarod test 3 times with an interval of 1 h between each test.

#### 2.2.3. Open Field Test (OFT)

The mice were allowed to adapt to their environment before the experiment. During the experiment, which lasted for a total of 5 min, the mice were positioned at the center of the OFT. The movement trials, total distance, central distance, and the number of crossings in the central zone were measured and recorded using the S-MART software (Pan Lab Co., Barcelona, Spain, version 3.0). After each experiment per animal, we wiped the area with 75% alcohol to remove the previous mouse’s smell.

### 2.3. Sample Collection and Tissue Preparation

A total of 7 mice in each group were randomly selected to collect feces. Each mouse was placed individually in an autoclaved cage in the morning following the last treatment. Fecal samples from mice were collected in sterile EP tubes on ice and stored at −80 °C.

All mice were anaesthetized with isoflurane. Whole blood was drawn from the heart. Following incubation at ambient temperature for 0.5 h, whole blood was centrifuged at 3000× *g* for 15 min at 4 °C. The collected supernatant was stored at −80 °C until subsequent use. A total of 7 mice in each group were randomly selected and in order to collect their fresh striatal and colonic tissues, they were perfused with sterile PBS. The tissue was collected on ice and stored at −80 °C for the determination of neurotransmitters and proteins.

The remaining 5 mice were used for staining experiments after whole brain tissue and colon tissue were collected. After perfusion with PBS and 4% paraformaldehyde (PFA), the colonic tissue was placed in 4% PFA for preservation. Whole brain tissue was fixed overnight in 4% PFA, followed by 24 h incubation in 20% sucrose solution and another 24 h incubation in 30% sucrose solution before embedding in OCT compound (SAURA, Shiogama, Japan). Whole brain tissue was stored at −80 °C for backup.

### 2.4. Determination of Neurotransmitter and Corresponding Metabolite Levels Using High-Performance Liquid Chromatography (HPLC)

The levels of striatal DA and serotonin (5-HT), as well as their corresponding metabolites, dihydroxyphenylacetic acid (DOPAC) and 5-hydroxyindoleacetic acid (5-HIAA), respectively, were measured using HPLC on a fluorescence detector (LC-20A, SHIMADZU, Kyoto, Japan). The Wonda Sil C18 Superb chromatographic column (4.6 mm × 150 mm, 5 μm, SHIMADZU, Japan) was used. In addition, methanol–citrate buffer (adjust pH to 5 with hydrochloric acid) served as the mobile phase. After the homogenization of the striatum in 0.1 m perchloric acid (1 mg, 10 μL), it was centrifugated at a rate of 12,000× *g* for 10 mins at 4 °C. Next, after collecting the supernatant, a filter with a pore size of 0.22 μm was employed for filtration, and then 10 μL of each sample was utilized for the detection procedure. The standard samples were dissolved in methanol. Standard curves were plotted using measurements from a range of dilutions of the mixed standard samples.

### 2.5. Immunofluorescence (IF) and Image Analysis

A cryostat microtome was used to slice the brain tissues of mice into 5 μm-thick sections (CM1950, Leica, Wetzlar, Germany). Sections containing the nigrostriatal fraction were collected. A total of 10 representative sections containing the major part of the substantia nigra from bregma 2.92 mm to 3.52 mm were selected for TH staining from the sections collected for each mouse. Briefly, following the repair of the slides with sodium citrate buffer (pH value, 6.0), brain slices were subjected to incubation in PBS with 0.2% *v*/*v* TritonX 100 for 10 min, followed by incubation in 5% *v*/*v* goat serum at 37 °C for 1 h. The samples were subjected to overnight incubation with the rabbit anti-TH (1:200, EP1532Y, Abcam, Cambridge, UK) primary antibody at 4 °C. After rewarming for 0.5 h, the primary antibody was detected with appropriate goat anti-rabbit antibody (1:500, A-11008, Thermo Fisher, Waltham, MA, USA) labeled with FITC. Nuclei were detected with DAPI solution (Solarbio, Beijing, China). The representative images were recorded with the aid of a fluorescence microscope (DFC7000T, Leica, Germany). Each group contained 5 animals. The TH-positive cells in substantia nigra from each group were counted with the ImageJ (NIH, Bethesda, MD, USA, version 1.53) software.

### 2.6. 16S rRNA Gene Sequencing for Analysis of the Gut Microbiome

The microbiota DNA was extracted using the TianGen TIANamp Bacteria DNA Kit. Barcoded primer sequences, 806R: GGACTACNNGGGTATCTAAT and 341F: CCTAYGGGRBGCASCAG, were utilized for the amplification of the V3-V4 hypervariable region of 16S rRNA, and the PCR-amplified products were purified following electrophoresis using 2% agarose gel. A library was constructed with The TruSeq^®^ DNA PCR-Free Sample Preparation Kit, after which it was quantitated using qPCR and Qubit. This was followed by library sequencing on the NovaSeq6000 (Illumina, San Diego, CA, USA) platform. The off-machine data were filtered; chimerism was eliminated, and other steps were undertaken to obtain effective tags. Based on the Uparse algorithm, effective tags were clustered into OTUs with an overall similarity of 97%. The SSUrRNA database of SILVA138 and the Mothur method were used for species annotation. The α-diversity and β-diversity indexes were computed utilizing QIIME (version 1.9.1), and the Wilcoxon test was conducted using R (version 2.15.3). An analytical tool, linear discriminant analysis (LDA) effect size (LEfSe), was utilized for identifying and interpreting high-dimensional biomarkers for detecting bacterial differences. For *p* < 0.05, the value of LDA > 4 indicated a significant enrichment.

### 2.7. Hematoxylin and Eosin (H&E) Staining

The colon of each mouse was paraffin-embedded and sliced into 5 μm-thick sections, which were subjected to the HE staining protocol and scored pathologically according to previously described criteria [27]. The degree of inflammation was defined as follows: 3, severe; 2, moderate; 1, mild; and 0, no inflammation. The extent of damage was defined as follows: 0, no damage; 1, damage to the mucosal layer; 2, damage to the submucosal layer; and 3, damage to the whole layer. Crypt disruption was scored as follows: 0, no disruption; 1, disruption to 1/3 of basal layer; 2, disruption to 2/3 of basal layer with an intact surface epithelium only; and 3, all epithelial basal layers were disrupted. Each of the above scores was multiplied by the proportion of colon involved (0–25% times 1, 26–50% times 2, 51–75% times 3, and 76–100% times 4) and finally, each score was summed to obtain the final pathology score.

### 2.8. Enzyme-Linked Immunosorbent Assay (ELISA)

The Interleukin-6 (IL-6) and tumor necrosis factor-α (TNF-α) levels in the sera of mice were measured with the aid of the corresponding kits (Solarbio Technology Co., Ltd., Beijing, China) following kit protocols. A standard protein curve was used to determine the target protein concentration. IL-6 and TNF-α concentrations were expressed in pg/L protein.

### 2.9. Western Blot Analysis

To extract protein from the striatum and colon, we added the RIPA lysis buffer with 1% phenylmethylsulphonyl fluoride (Beyotime, Shanghai, China) to the ground tissue. Subsequently, we homogenized the samples and centrifugated them at a rate of 12,000× *g* for 15 min at 4 °C. Thereafter, the BCA Protein Assay Kit (Beyotime, Shanghai, China) was employed to measure the supernatant’s protein concentration after centrifugation. SDS-PAGE (sodium dodecyl sulfate-polyacrylamide gel electrophoresis) was employed to extract the denatured protein sample before it was transferred onto a polyvinylidene fluoride (PVDF) membrane (Millipore, Burlington, MA, USA). Next, the membranes were blocked following protein transfer with 5% skim milk before being treated for an entire night with the specific primary antibodies at 4 °C. Specifically, the primary antibodies below were utilized in the experiments: rabbit anti-TH antibody (1:1000, 25859-1-AP, Proteintech, Rosemont, IL, USA), rabbit anti-GFAP antibody (1:1000, K106966P, Solarbio), rabbit anti-Iba-1 antibody (1:1000, 10904-1-AP, Proteintech), rabbit anti-occludin antibody (1:1000, 27260-1-AP, Proteintech), rabbit anti-ZO-1 (1:1000, 21773-1-AP, Proteintech), rabbit anti-TLR4 antibody (1:1000, 19811-1-AP, Proteintech), rabbit anti-MyD88 antibody (1:1000, ab40676, Abcam), rabbit anti-IL-6 antibody (1:1000, K009385P, Solarbio), rabbit anti-TNF-α antibody (1:1000, 17590-1-AP, Proteintech), rabbit anti-NF-κB antibody (1:1000, #8242, Cell Signaling Technology, Danvers, MA, USA), rabbit anti-β-actin antibody (1:1000, 20536-1-AP, Proteintech), and rabbit anti-GAPDH antibody (1:1000, 10494-1-AP, Proteintech). The primary antibody was recognized using an HRP-conjugated anti-rabbit antibody (1:5000, BA1054, Boster, San Mateo, CA, USA). The chemiluminescence solution (Millipore, Billerica, MA, USA) was dripped on the PVDF membrane and exposed to an automatic gel imaging analyzer (Bio-Rad, Hercules, CA, USA). Lastly, ImageJ was applied to measure the protein band density.

### 2.10. Statistical Analysis

The GraphPad software (version 8.0.2), was employed for the analyses of statistical data. A one-way analysis of variance (ANOVA) followed by Tukey’s post-hoc test was conducted for detecting differences. The Kruskal–Wallis test, followed by Dunn’s multiple comparison test (α < 0.05) or the Wilcoxon rank sum test, was conducted to make a comparison among three groups of specific taxa of microbiota. All data were presented as mean ± standard error of the mean (SEM). A value of *p* < 0.05 indicated the significance criterion.

## 3. Results

### 3.1. NaB Enhances Motor Functions and Exploratory Behaviors in Mice with MPTP-Induced PD

We tested the effects of NaB on the motor function in mice with PD using poles and rotarods. In comparison to the control mice, the MPTP-induced mice exhibited severe locomotor impairment, as shown by a considerably longer pole test time and a considerably shorter rotarod test time (*p* < 0.001). In contrast, the MPTP + NaB mice exhibited significantly better performance in the pole test (*p* < 0.001) and rotarod test (*p* < 0.001) relative to the MPTP mice (Figure 1B,C). Subsequently, we assessed the effect of NaB on locomotor abilities and exploratory behaviors based on the OFT by measuring the paths traversed by the mice (Figure 1D). A lesser total distance (*p* < 0.001), a smaller central distance (*p* < 0.001), and fewer crossings in the central zone (*p* < 0.001) were found in MPTP mice relative to the control mice. The performance in terms of the total distance (*p* < 0.001), central distance (*p* < 0.001), and the total number of crossings in the central zone (Figure 1E–G) (*p* < 0.001) was better for the MPTP + NaB mice relative to the MPTP mice. These results demonstrated that NaB could improve locomotor functions and exploratory behaviors in the PD mice.

### 3.2. NaB Elevates the Levels of DA and 5-HT in the Striatum of MPTP-Induced PD Mice

To ascertain whether or not NaB had a neuroprotective function, we used fluorescence detection after HPLC to measure the levels of striatal neurotransmitters, as well as their metabolites. A 31.5% decrease in DA levels in the striatum of MPTP mice (*p* < 0.01) was observed. In contrast, DA levels in the striatum increased remarkably by 49.2% in the MPTP + NaB mice (*p* < 0.01) relative to the MPTP mice (Figure 2A). Additionally, 5-HT levels decreased by 34.6% in the MPTP mice (*p* < 0.01), whereas these increased by 49.2% in the MPTP + NaB mice (*p* < 0.01) (Figure 2B). The MPTP mice tended to have lower DOPAC and 5-HIAA levels than the normal mice but these differences were statistically insignificant. The striatal levels of DOPAC and 5-HIAA were elevated remarkably in the MPTP + NaB group by 38.7% and 32.0%, respectively, compared to the MPTP group (*p* < 0.05 for both) (Figure 2C,D).

### 3.3. NaB Enhances the Number of Dopaminergic Neurons and the Level of TH in Mice with MPTP-Induced PD

We conducted an IF analysis on the PD mice to determine if NaB protected the TH+ dopaminergic neurons. Striatal TH protein expression was examined using Western blotting. As shown using IF analysis, a remarkable decrease in TH+ dopaminergic neurons of 51.2% was seen in the PD mice (*p* < 0.01) compared to the control mice but NaB exposure inhibited the decline by 83.0% compared to the MPTP group (*p* < 0.05) (Figure 3A,B). Consistently, the Western blotting demonstrated that the TH levels in the MPTP mice decreased remarkably by 47.5% (*p* < 0.01) compared to the control mice, while the NaB treatment increased the TH expression significantly (*p* < 0.01) by 93.0% compared to MPTP alone (Figure 3C,D). The above results suggested that NaB performed a neuroprotective function in the PD mice by increasing striatal TH expression and restoring dopaminergic neuron loss.

### 3.4. NaB in Mice with MPTP-Induced PD Attenuates Microglia and Astrocyte Activation

Neuroinflammation in neurodegenerative diseases, including PD, may be associated with astrocyte and microglial activation [28]. To assess whether microglia and astrocytes were activated, Western blotting was conducted for the detection of protein markers, Iba-1 and GFAP, in microglia and astrocytes, respectively. Additionally, in the striatum of the PD mice, Iba-1 levels were substantially higher by 102.2% (*p* < 0.001), while the NaB treatment in the mice with MPTP-induced PD decreased its expression by 45.4% compared to the MPTP-treated mice (*p* < 0.001) (Figure 4A). As well, the MPTP mice displayed a 53.6% decrease in GFAP expression compared to the control mice in the striatum (*p* < 0.001). The treatment with NaB decreased GFAP expression (*p* < 0.01) by 26.2% in the MPTP mice (Figure 4B,C). Thus, the NaB treatment inhibited glial cell activation, leading to reduced neuroinflammation.

### 3.5. NaB Modifies Dysbiosis of the Gut Microbiome in Mice with PD Caused by MPTP

To assess the ameliorative impact of NaB exposure on gut microbiome dysbiosis in PD mice, a 16S rRNA seq-analysis of fecal gut microbiota was conducted in each mouse group. First, an α-diversity analysis was performed to assess bacterial diversity and richness. The MPTP group exhibited significantly higher Shannon and Simpson indexes relative to the controls. The NaB mice showed a downward trend relative to the MPTP group but this was statistically insignificant (Figure 5A,B). Furthermore, β-diversity was assessed based on weighted UniFrac distances and Bray–Curtis distances to determine the similarity in microbiota composition across these three groups (Figure 5C,D). A significant difference in the gut microbiota was observed, indicating substantial variations in the overall structure of the gut microbiota across the control, MPTP, and MPTP + NaB mice. Moreover, the results demonstrated that the control and MPTP mice were separated, with the NaB treatment group located in between the two groups, suggesting that the NaB treatment shifted the overall MPTP-damaged gut microbiota composition towards that of the control group. To further clarify the key bacteria causing the occurrence of PD, the relative abundances of different taxa of microorganisms were compared (Figure 5E,F). A dramatic change in microbiota composition was observed. In terms of the phylum level, Verrucomicrobiota and Proteobacteria were elevated in the PD mice (*p* < 0.001, *p* < 0.05, respectively) but the NaB treatment significantly decreased their abundances (*p* > 0.05, *p* > 0.05, respectively). In terms of the family level, the MPTP mice recorded an increased abundance of Bacteroidaceae (*p* < 0.001), along with a reduction in Erysipelotrichaceae abundance (*p* < 0.05). However, the abundance of Bacteroidaceae was lowered in the NaB group (*p* < 0.05), whereas that of Erysipelotrichaceae was elevated considerably (*p* > 0.05). In terms of the bacterial genera, the MPTP mice had higher relative abundances of *Bacteroides*, *Parabacteroides*, *Akkermansia,* and *Lactobacillus* (*p* < 0.01, *p* < 0.01, *p* < 0.01, *p* < 0.05, respectively), while those of *Dubosiella* and *Odoribacter* in the MPTP group were lowered (*p* < 0.05, *p* < 0.01, respectively). However, the relative abundance of *Parabacteroides*, *Bacteroides*, *Akkermansia,* and *Lactobacillus* decreased after NaB administration (*p* > 0.05, *p* < 0.05, *p* > 0.05, *p* < 0.05, correspondingly), whereas that of *Dubosiella* and *Odoribacter* was elevated substantially (*p* > 0.05, *p* < 0.05, respectively) (Figure 5G). To assess the specific bacterial taxa associated with PD, we determined the mice gut microbiota composition across the three groups using LEfSe based on relative abundances, along with the significant changes (Figure 5H,I). The data suggested that the MPTP-induced mice had gut microbial dysbiosis, and that the NaB treatment improved gut dysbiosis in these animals by modulating microbiota composition.

### 3.6. NaB Ameliorates Colonic Epithelial Barrier Damage in Mice with MPTP-Induced PD

To verify the impact of NaB on colonic pathology, HE staining on mouse colonic sections was performed to observe inflammation. The MPTP group showed more severe colonic inflammation relative to the control group. The NaB group exhibited lower colonic inflammation than the MPTP group (Figure 6A). Moreover, the MPTP mice showed significantly higher HE histological scores (*p* < 0.01), whereas the NaB-treated mice showed significantly lower scores (*p* < 0.05) (Figure 6B). Western blotting was conducted to measure the expression profiles of ZO-1 and occludin to assess the influence of NaB administration on the colonic barrier (Figure 6C). The expression levels of ZO-1 and occludin were reduced remarkably by 32.1% and 55.8%, respectively, in the colon of the MPTP mice (*p* < 0.05 and *p* < 0.01, respectively), whereas in the MPTP + NaB mice, the occludin and ZO-1 levels increased by 100.5% and 49.2% (*p* < 0.01 and *p* < 0.05, respectively) (Figure 6D,E). Taken together, NaB could restore colonic barrier damage in the MPTP mice.

### 3.7. NaB Attenuates Neuroinflammation and Intestinal Inflammation by Inhibiting the TLR4/MyD88/NF-κB Signaling Pathway

The gut microbiota in the PD mice was found to contain an abundance of pro-inflammatory bacteria based on the 16S rRNA sequencing analysis. The intestinal permeability in the PD mice was also increased. Therefore, we first detected inflammatory factor levels in the colon, serum, and striatum of the mice, followed by the TLR4/MyD88/NF-κB pathway expression in the colon and striatum. The ELISA findings illustrated that the IL-6 and TNF-α levels in the serum were higher by 193.6% and 150.5%, respectively, in MPTP mice (*p* < 0.001 for both), whereas these levels decreased correspondingly by 64.2% and 50.0% in the MPTP + NaB mice (*p* < 0.001 for both) (Figure 7A,B). The Western blotting illustrated a substantial decline in the levels of TNF-α and IL-6 of 78.7% and 55.1%, respectively, in the striatum of the MPTP mice (*p* < 0.01, *p* < 0.05), while the NaB treatment considerably lowered the corresponding expression levels by 41.0% and 37.9% (*p* < 0.01, *p* < 0.05, respectively) (Figure 7C–E). The colonic TNF-α and IL-6 levels increased remarkably by 88.2% and 83.6% in the MPTP mice (*p* < 0.001, *p* < 0.05, respectively), whereas these levels correspondingly decreased by 45.0% and 42.0% in the MPTP + NaB mice (*p* < 0.001, *p* < 0.05, respectively) (Figure 7F–H).

The Western blot results for the signaling pathway proteins indicated that the TLR4, MyD88, and NF-κB levels increased significantly in the striatum by 78.4%, 75.8%, and 91.6%, respectively, in the PD mice (*p* < 0.05, *p* < 0.001, *p* < 0.01, respectively), whereas these levels declined significantly by 45.4%, 44.1%, and 45.9% in the MPTP + NaB mice (*p* < 0.05, *p* < 0.001, *p* < 0.01, respectively) (Figure 7I–L). Similarly, TLR4, MyD88, and NF-B protein expression decreased nearly 70%, 94.7%, and 94.2% in the colons of the PD group mice, respectively (*p* < 0.01, *p* < 0.05, *p* < 0.05, respectively). In contrast, NaB exposure substantially decreased these protein expression levels by 41.2%, 67.0%, and 48.0% (*p* < 0.01, *p* < 0.01, *p* < 0.05, respectively) (Figure 7M–P). Thus, NaB reduced intestinal inflammation and neuroinflammation by suppressing the activities of the TLR4/MyD88/NF-κB signaling pathway.

## 4. Discussion

Growing evidence suggests the association of gut microbiota dysbiosis with PD occurrence, development, and progression [29,30]. Altering the gut microbiota composition is a viable target for diagnosing and treating PD [31,32]. The NaB treatment had no significant effect on normal animals, especially on gut microbiota [22], intestinal inflammation [33], neuroinflammation [34], and motor function [35]. However, NaB plays an important role in the treatment of disease. Notably, NaB exerts anti-apoptotic, anti-inflammatory, and neuroprotective effects on mice with PD [36,37,38] but the exact mechanism underlying these effects of NaB in PD remains unclear. According to the results of this study, NaB improved gut microbial dysbiosis and inhibited intestinal inflammation and neuroinflammation, resulting in neuroprotective effects, which involved the TLR4/MyD88/NF-κB pathway. Figure 8 depicts the possible processes via which NaB treated MPTP-induced PD mice.

PD is characterized by lowered dopamine levels in the striatum due to the death and degeneration of dopaminergic neurons. TH, a rate-limiting enzyme, catalyzes the tyrosine synthesis of L-dopa and mediates DA synthesis. TH downregulation decreases DA production, in turn leading to PD [39]. The NaB treatment reversed the loss of TH-positive cells in the substantia nigra of the PD mice, as well as the decline in DA and TH levels in the striatum, consistent with previous findings [23,25]. The content of 5-HT reportedly decreases in mouse models of PD [13,40]. The 5-HT and DA neurotransmitters are involved in regulating movement and balance. The motor symptoms of PD may result from decreased dopamine and serotonin levels [41,42]. The NaB treatment ameliorates motor symptoms in a *Drosophila* rotenone-induced PD model [43]. We conducted behavioral experiments on mice and found that NaB treatment attenuated MPTP-induced motor symptoms. According to these findings, NaB was neuroprotective in MPTP-induced PD mice, thus reducing motor impairments.

Considering the effects of NaB on the gut microbiota as well as the microbiota–gut–brain axis in PD, a 16S rRNA analysis of the gut microbiota was performed. Consistent with other MPTP-induced PD models [44,45], the community structure of the microbiota in our animal model changed significantly. The MPTP treatment increased the gut microbiota’s α-diversity. Although the NaB treatment reduced the α-diversity, the difference was statistically insignificant, suggesting that the sample size needs to be increased in the future. The β-diversity analysis demonstrated that the microbial community morphologies of the NaB-treated and control groups were similar. Furthermore, the 16S rRNA data showed abnormal microbial compositions at different taxonomic levels. A remarkable elevation in the abundance of Proteobacteria was noted in the PD mice, consistent with previous reports on PD patients’ gut microbiota [46,47]. The Proteobacteria phylum is a marker of dysbiosis in the gut microbiome, and Proteobacteria with pro-inflammatory characteristics are often overexpressed in gastrointestinal and immune diseases [48]. The abundance of *Bacteroides* also increased significantly in PD mice and is known to correlate with an increase in plasma TNF-α and interferon-γ levels in patients with PD [49]. Moreover, *Bacteroides fragilis* belongs to the genus *Bacteroides*, responsible for systemic inflammation owing to its export of the highly inflammatory zinc metalloprotease, fragilysin, in addition to the massive synthesis of the highly neurotoxic *Bacteroides fragilis*-lipopolysaccharide (LPS) [50]. Both mice and patients with PD exhibit increased levels of the genus *Akkermansia* [51,52]. The abundance of *Akkermansia* was found to be significantly elevated in mice with PD consistently in our study. The Gram-negative bacteria, *Akkermansia*, can degrade mucin, cause intestinal inflammation and permeability, and increase endotoxemia and systemic inflammation, eventually promoting neuropathology [53,54,55]. Notably, the NaB treatment inhibited the abnormal increase in the abundances of Proteobacteria, *Bacteroides*, and *Akkermansia*, which might exert beneficial effects on the PD mice. Interestingly, the PD mice had a higher abundance of *Lactobacillus*, consistent with findings in patients with Crohn’s disease and PD [56,57]. *Lactobacillus* is generally considered a beneficial bacteria; however, whether *Lactobacillus* can affect PD or simply thrive in pro-inflammatory environments remains unclear [58]. The PD mice showed a significant reduction in *Odoribacter* abundance. *Odoribacter* is a beneficial bacteria known for producing SCFAs and reducing inflammation [59]. The abundance of *Odoribacter* was significantly elevated after the NaB treatment. Accordingly, MPTP-induced dysbiosis of the gut microbiota might contribute to an inflammatory state, while NaB modulated the microbial diversity and gut microbiota’s composition in the PD mice to decrease harmful pathogen abundance while increasing the abundance of beneficial bacteria. We reasonably concluded that the gut microbiota may be a promising treatment target for NaB in treating PD.

An abnormal increase in pro-inflammatory and other harmful bacteria caused by disorders of the gut microbiota can trigger gut inflammation [46]. Herein, we observed a remarkable increase in the colonic HE pathology score in the PD mice, in addition to the elevated expression levels of colonic inflammatory factors, TNF-α and IL-6. All of these results indicated an intense intestinal inflammatory response in the PD mice, and the NaB treatment significantly reduced intestinal inflammation. According to the popular “leaky gut” theory, gut inflammation increases intestinal permeability [60]. PD patients and animals have both been found to have increased intestinal permeability in previous studies [61,62]. The preservation of the intestinal barrier’s integrity relies on the tight junctions between intestinal epithelial cells and their molecular structure including the cytoplasmic and transmembrane proteins, ZO, and occludin [63]. Our results indicated increased intestinal permeability in the PD mice, as ZO-1 and occludin levels were decreased. In contrast, NaB exposure contributed to upregulated expression of these proteins and restored intestinal barrier function in the PD mice. Due to increased intestinal permeability, microbial toxins and pro-inflammatory mediators produced as a result of gut dysbiosis can leak into the circulation causing systemic inflammation, in turn destroying the blood–brain barrier and causing neuroinflammation [12,64]. The TNF-α and IL-6 levels were substantially elevated in the serum and striatum, and the NaB treatment suppressed their elevation. Thus, treatment with NaB may reduce inflammation and disruption of the intestinal barrier caused by gut microbial dysbiosis, thereby blocking the spread of pro-inflammatory factors and harmful gut microbial metabolites throughout the body and reducing MPTP-induced neuroinflammation.

Several studies have confirmed that Toll-like receptors, specifically TLR4, play an instrumental role in intestinal homeostasis [65]. Accumulating evidence suggests that altered TLR4 signaling contributes to PD pathology. For instance, patients with PD show elevated TLR4 protein levels in their colonic biopsies, circulating monocytes, as well as postmortem caudate and putamen samples [66,67]. The MPTP-induced PD mice showed attenuation in intestinal inflammation and neuroinflammation upon the knocking out of TLR4 [67]. The TLR4/MyD88/NF-κB pathway is well recognized for its role in the inflammatory signaling pathway [68]. Herein, TLR4/MyD88/NF-B pathway activation was observed in the colon and striatum of the PD mice, possibly due to increased Proteobacteria abundance. Inflammatory LPS is produced by abnormally high Proteobacteria, and TLR4 recognizes elevated inflammatory LPS [69], in turn activating the MyD88-dependent NF-κB signaling pathway. TNF-α and IL-6 are downstream products of NF-KB, so when the TLR4 pathway is activated, they are produced more readily [70,71]. Herein, elevated TNF-α and IL-6 levels were demonstrated in both the colon and striatum of the PD mice. The NaB treatment, however, inhibited TLR4/MyD88/NF-B pathway activation in the colon and striatum. Interestingly, activation of the TLR4/MyD88/NF-κB pathway can induce astrocyte and microglial activation [72]. The release of pro-inflammatory factors and neurotoxins by activated glial cells leads to neuroinflammation, neuronal damage, and neurodegeneration [73]. The NaB treatment inhibited microglial activation and astrocyte activity, thus protecting against neuroinflammation and neurodegeneration. Based on these results, the NaB treatment reduced MPTP-induced inflammation in the gut–brain axis of the PD mice, which might be associated with the TLR4/MyD88/NF-B signaling pathway.

In conclusion, the neuroprotective effect of NaB on MPTP-induced PD mice may be attributed to the recovery of the composition of the gut microbiota after NaB treatment. Additionally, the NaB treatment modulated inflammation throughout the gut–brain axis by inhibiting the TLR4/MyD88/NF-kB signaling pathway, a protective mechanism underlying its action. As a result, NaB seems to be a potential drug that may be effective at treating PD. However, further investigation is needed to demonstrate regulation of the gut microbiota structure by NaB. Moreover, this study was limited to animals and its applicability to clinical populations remains a major challenge that needs to be addressed in the future.

## Figures and Tables

**Figure 1 nutrients-15-00930-f001:**
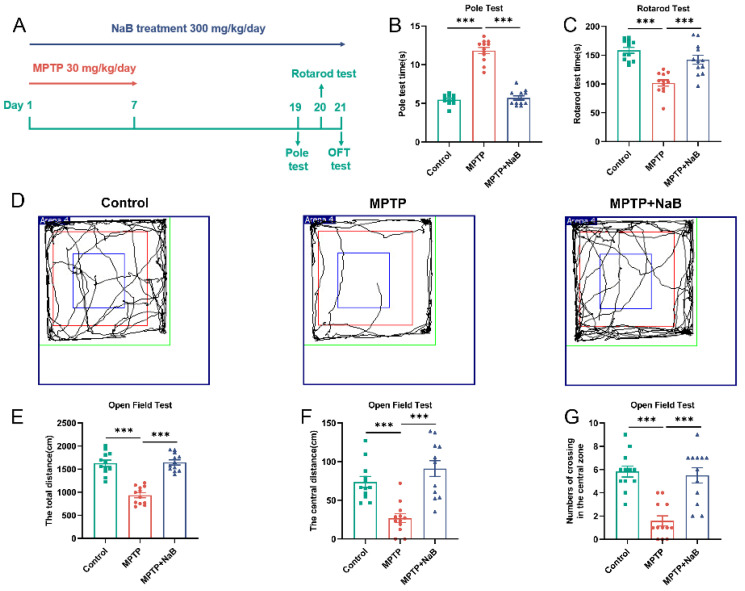
Sodium butyrate (NaB) improves motor functions and exploratory behaviors in mice with 1-methyl-4-phenyl-1,2,3,6-tetrahydropyridine (MPTP)-induced Parkinson’s disease (PD). (**A**) A diagrammatic representation of the treatment given to animals. (**B**) Pole test. (**C**) Rotarod test. (**D**) Representative images from open field test (OFT) movement trials. The (**E**) total distance, (**F**) central distance, (**G**) number of crossings in the central zone of the OFT. *n* = 12 mice per group; *** *p* < 0.001; data are expressed as means ± SEM.

**Figure 2 nutrients-15-00930-f002:**
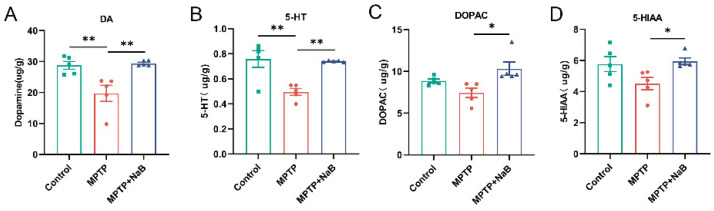
Sodium butyrate (NaB) increases dopamine (DA) and serotonin (5-HT) levels in the striatum of mice with 1-methyl-4-phenyl-1,2,3,6-tetrahydropyridine (MPTP)-induced Parkinson’s disease (PD). Levels of the neurotransmitters (**A**) DA, (**B**) 5-HT, and their metabolites (**C**) dihydroxyphenylacetic acid (DOPAC), and (**D**) 5-hydroxyindoleacetic acid (5-HIAA) in the striatum. *n* = 5 mice per group; * *p* < 0.05, ** *p* < 0.01; data are presented as means ± SEM.

**Figure 3 nutrients-15-00930-f003:**
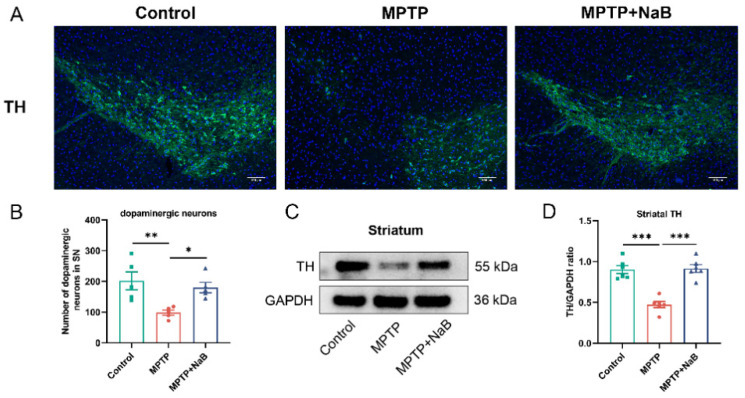
Sodium butyrate (NaB) enhances the number of dopaminergic neurons and tyrosine hydroxylase (TH) expression in Parkinson’s disease (PD) mice. (**A**) Images showing immunofluorescence staining in the substantia nigra for markers TH (green) of dopaminergic neurons and nuclei DAPI (blue). The scale bar measures 100 μm. (**B**) Quantification of TH+ dopaminergic neurons in the substantia nigra using ImageJ. (**C**) Representative Western blot that indicated TH expression in the striatum. (**D**) The intensities of the TH bands were quantified using ImageJ and normalized to GAPDH. *n* = 5 mice for each group; * *p* < 0.05, ** *p* < 0.01, *** *p* < 0.001; data are expressed as means ± SEM.

**Figure 4 nutrients-15-00930-f004:**
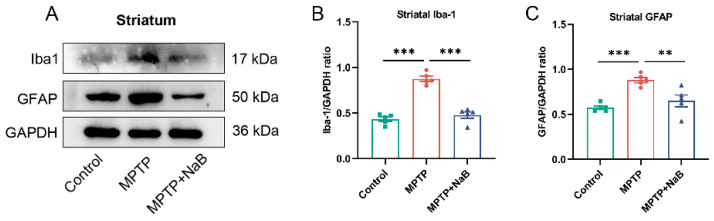
Sodium butyrate (NaB) inhibits the activation of microglia and astrocytes in 1-methyl-4-phenyl-1,2,3,6-tetrahydropyridine (MPTP)-induced Parkinson’s disease (PD) mice. (**A**) Representative Western blot for Iba-1 and GFAP expression in the striatum. (**B**,**C**) The density analysis for Iba-1 and GFAP bands in the striatum. Band intensities were normalized to GAPDH for quantification. *n* = 5 mice per group; ** *p* < 0.01, *** *p* < 0.01; data are presented as means ± SEM.

**Figure 5 nutrients-15-00930-f005:**
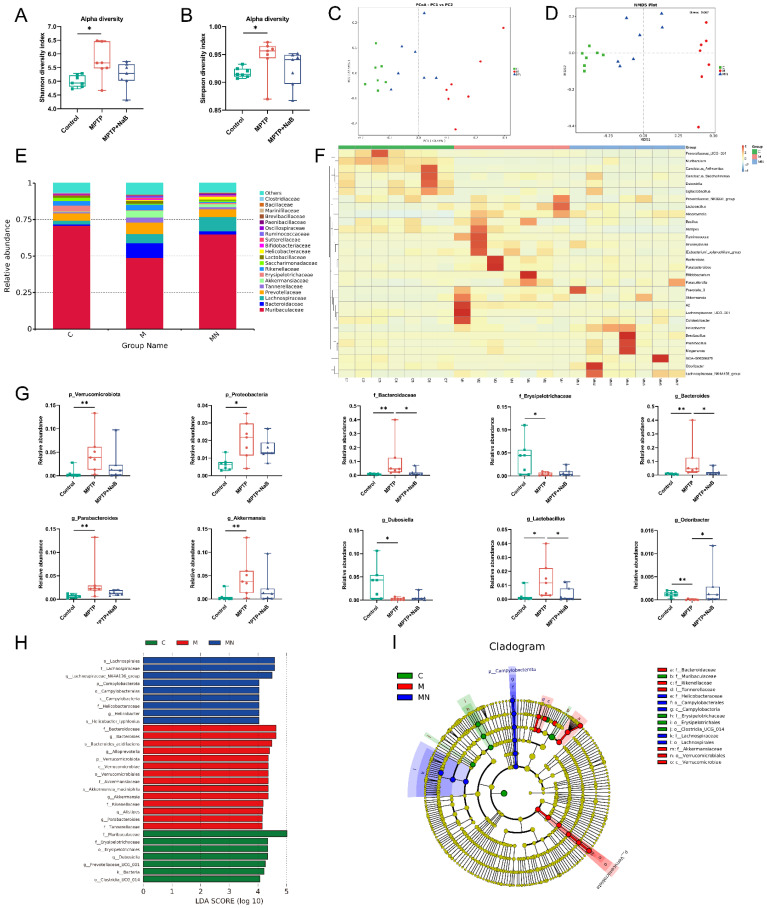
Sodium butyrate (NaB) alters gut microbiome dysbiosis in 1-methyl-4-phenyl-1,2,3,6-tetrahydropyridine (MPTP)-induced Parkinson’s disease (PD) mice. α-diversity indexes of gut microbiota including the (**A**) Shannon and (**B**) Simpson indexes. (**C**) Principal coordinate analysis (PCoA) plots based on weighted UniFrac distances show β-diversity indexes of the gut microbiome. (**D**) β-diversity indexes of the gut microbiome are displayed using nonmetric multidimensional scaling (NMDS) based on Bray–Curtis distances. (**E**) The relative abundance of the gut microbiome in the three groups is shown by bar plots at the family level. (**F**) The results of the heatmap analysis demonstrate the relative abundance of gut microbiota at a genus level for each of the three groups. (**G**) The relative abundances of p_Verrucomicrobiota, p_Proteobacteria, f_Bacteroidaceae, f_Erysipelotrichaceae, g_Bacteroides, g_Parabacteroides, g_Akkermansia, g_Dubosiella, g_Lactobacillus, and g_Odoribacter across the three different groups. (**H**) Histograms and (**I**) cladograms of bacterial taxa across the three groups (cutoffs were linear discriminant analysis (LDA) >4 and *p*-value < 0.05). Green, red, and blue shading on the cladogram indicate bacteria with higher abundances in the control, MPTP, and MPTP + NaB mice, respectively. C: control, M: MPTP, MN: MPTP + NaB. p: phylum; c: class; o: order; f: family; g: genus; s: species. *n* = 7 per group, * *p* < 0.05, ** *p* < 0.01. Data are expressed as means ± SEM.

**Figure 6 nutrients-15-00930-f006:**
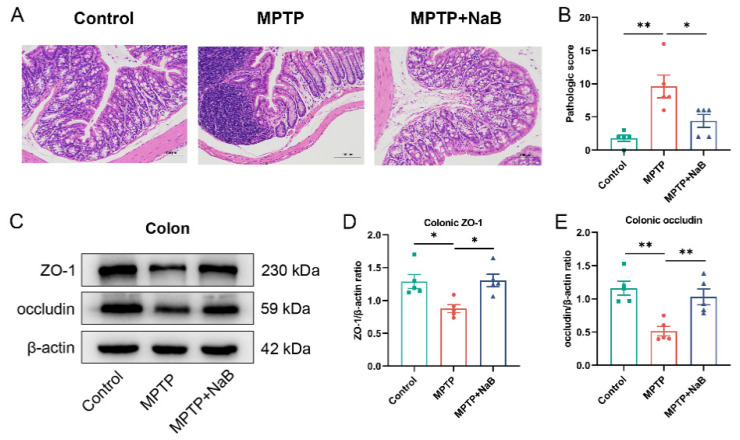
Sodium butyrate (NaB) ameliorates the damage to the colonic epithelial barrier in mice with 1-methyl-4-phenyl-1,2,3,6-tetrahydropyridine (MPTP)-induced Parkinson’s disease (PD). (**A**) Hematoxylin and Eosin (H&E) staining of colonic tissues. (**B**) Histology scoring for colonic sections based on HE staining. (**C**) Representative Western blotting for colonic ZO-1 and occludin expressions. (**D**,**E**) Band intensities of ZO-1 and occludin were quantified using ImageJ and normalized against β-actin levels. *n* = 5 mice per group; * *p* < 0.05, ** *p* < 0.01; data are presented as means ±SEM.

**Figure 7 nutrients-15-00930-f007:**
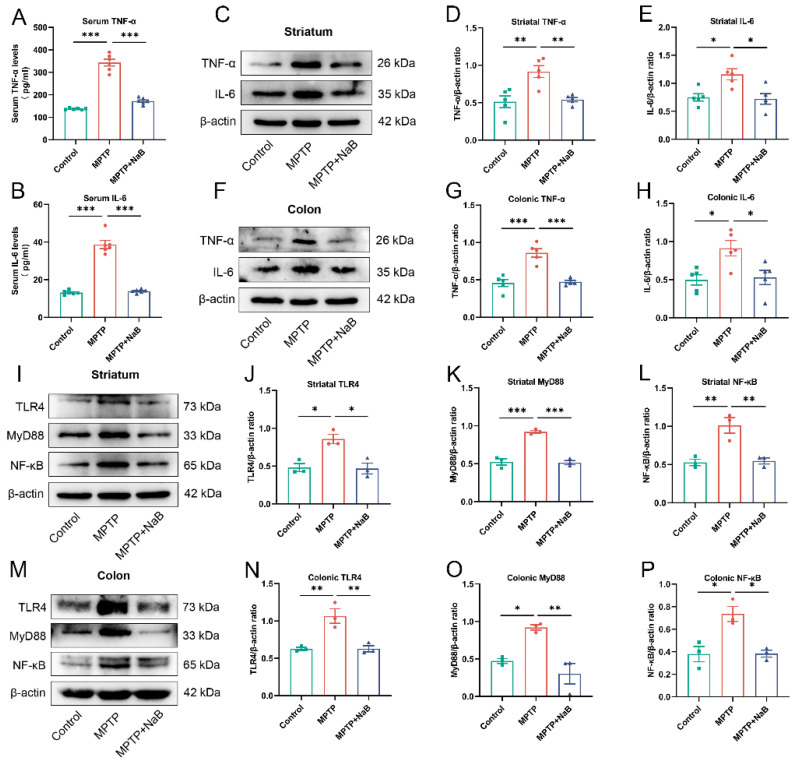
Neuroinflammation and intestinal inflammation were attenuated following Sodium butyrate (NaB) treatment due to the inhibition of the TLR4/MyD88/NF-B signaling pathway. (**A**,**B**) The levels of serum inflammatory cytokines, TNF-α, and IL-6. (**C**,**F**) Representative Western blots for striatal and colonic IL-6 and TNF-α expression. (**D**,**E**,**G**,**H**) Band intensities of TNF-α and IL-6 in the striatum and colon were quantified using ImageJ and normalized to β-actin. (**I**,**M**) TLR4, MyD88, and NF-B expression levels in the striatum and colon are shown in a representative Western blot. (**J**–**L**,**N**–**P**) The band density analysis for TLR4, MyD88, and NF-κB proteins in the striatum and colon. Band intensities were normalized to β-actin for quantification. For (**A**,**B**), *n* = 6. For (**D**,**E**,**G**,**H**), *n* = 5. For (**J**–**L**,**N**–**P**), *n* = 3. * *p* < 0.5, ** *p* < 0.01, *** *p* < 0.001; data are presented as means ± SEM.

**Figure 8 nutrients-15-00930-f008:**
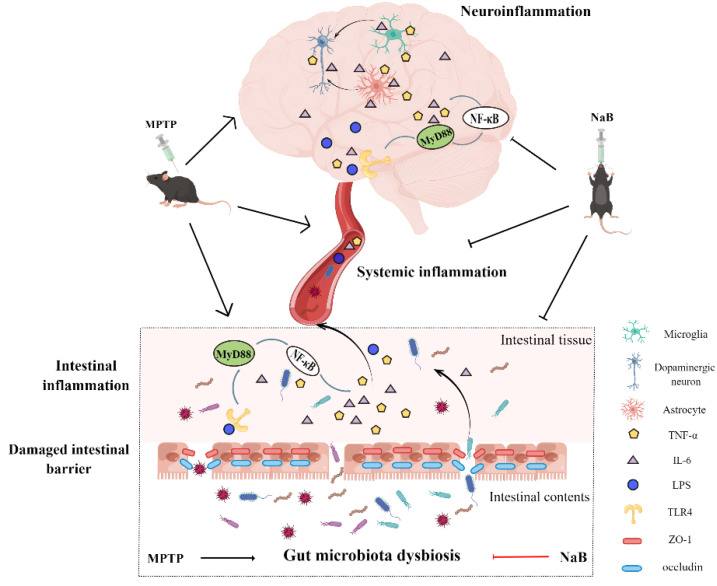
Overview of the protective mechanisms of Sodium butyrate (NaB) treatment in mice with 1-methyl-4-phenyl-1,2,3,6-tetrahydropyridine (MPTP)-induced Parkinson’s disease (PD). NaB treatment attenuates MPTP-induced gut microbiota dysbiosis, thereby inhibiting the production of intestinal pro-inflammatory factors and toxins along with decreasing intestinal permeability. Therefore, fewer pro-inflammatory factors, toxins, and microbes enter systemic circulation through the broken gut barrier. Consequently, the amount of pro-inflammatory factors and toxins entering the brain also reduces, thus inhibiting glial cell activation and neuroinflammation, ultimately leading to a reduction in dopaminergic neuronal death. Further, NaB treatment inhibits the activation of the TLR4/MyD88/NF-B signaling pathway in the gut and brain, a potential underlying molecular mechanism.

## Data Availability

Our data has other research purposes and is not readily available to the public.

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
