# Peer review of "Neuroprotective Effects of Sodium Butyrate by Restoring Gut Microbiota and Inhibiting TLR4 Signaling in Mice with MPTP-Induced Parkinson’s Disease"

_nutrients, 2023, doi:10.3390/nu15040930_

Round 1
Reviewer 1 Report
Comments for authors
Nutrients – Manuscript ID: nutrients-2157881 – “Neuroprotective Effects of Sodium Butyrate by Restoring Gut Microbiota and Inhibiting TLR4 Signaling on Mice with MPTP-Induced Parkinson’s Disease”, by Tong-tong Guo, Zheng Zhang, Yan Sun, Rui-yang Zhu, Fei-xia Wang, Lian-ju Ma, Lin Jiang, Han-Deng Liu.
The authors conducted an interesting and comprehensive study on potential neuroprotective effects of sodium butyrate (NaB) on mice with MPTP-induced Parkinson’s disease. Motor behavioral tests were performed and cellular and molecular analysis were done. The composition of the gut microbiota was also analyzed. Authors shown that NaB improved the motor functioning of PD mice, increased striatal neurotransmitter levels, and reduced the death of dopaminergic neurons. 16S rRNA sequencing analysis revealed that NaB restored the gut microbial dysbiosis. NaB also attenuated the intestinal barrier’s disruption and reduced serum, colon, and striatal pro-inflammatory cytokines, along with inhibiting the overactivation of glial cells. Authors concluded that their results proved that NaB had a neuroprotective impact on PD mice, likely linked to its regulation of gut microbiota to inhibit gut-brain axis inflammation.
The study contains some interesting data, which could have potential clinical implications for the therapeutic strategies of PD. There are, however, some inconsistencies in the methods that they need further explanation.
The first is about mice groups. In my opinion, a control group is missing to validate the NaB’s neuroprotective effects suppressing gut-brain axis inflammation. Authors should add a control + NaB group to assess the NaB action on gut microbiota and on mice’s motor behavior, on striatal dopamine and serotonin neurotransmitters levels and on nigral dopaminergic neurons. Then in this case, another statistical test should be performed, a two-ways (group and treatment) ANOVA followed by a post-hoc test.
Secondly, the distribution of animals for the different in vitro, histological, cellular and molecular analyzes is not clear. The authors mention the fact that the study involves 12 animals per group. This seems to be the case for in vivo analyzes. However, it’s not clear why 5 animals were used for in vitro brain immunohistochemistry analysis, 5 animals for striatal DA and 5-HT levels assessments, 5 for striatal Iba-1 and GFAP expression analysis and 5 for microbiota composition analysis. It’s probable not same animals, but this needs to be clarified.
Then, I have some minor points:
1. Authors mentioned in the introduction chapter, page 2 of 20, line 76 “We assessed the effects of NaB on dyskinesia, decline in striatal tyrosine hydroxylase (TH) levels, dopaminergic neuron loss, and reduction in striatal neurotransmitter expression in PD mice.”. Dyskinesia in PD are levodopa-induced dyskinesia. What do the authors mean? Dyskinesias are not studied in this article.
2. Immunofluorescence (IF) staining in the substantia nigra nuclei DAPI is not very relevant especially for the chosen images.
3. Authors should explain how they analyzed IF results with ImageJ (number of neurons on which substantia nigra sections, anterior, median, posterior SN, total SN, etc…). They will be able to do the same for other analyses also using ImageJ.
Finally, if this article is updated during the revision of the manuscript, I will recommend its publication.

Reviewer 2 Report
see attached file

Round 2
Reviewer 1 Report
The authors submitted a carefully-prepared revision, which satisfactorily addressed the remaining concerns.
Therefore, I recommend publication.